# Hepatic-Specific FGF21 Knockout Abrogates Ovariectomy-Induced Obesity by Reversing Corticosterone Production

**DOI:** 10.3390/ijms241914922

**Published:** 2023-10-05

**Authors:** Jiayu Xu, Xinyu Shao, Haozhe Zeng, Chengxi Wang, Jiayi Li, Xiaoqin Peng, Yong Zhuo, Lun Hua, Fengyan Meng, Xingfa Han

**Affiliations:** 1College of Life Science, Sichuan Agricultural University, Ya’an 625014, China; xujiayu0117@163.com (J.X.); 13541140468@163.com (X.S.); 17858424481@163.com (H.Z.); 17808353302@163.com (C.W.); 19938295065@163.com (J.L.); 15213246358@163.com (X.P.); mfy0407@126.com (F.M.); 2Institute of Animal Nutrition, Sichuan Agricultural University, Chengdu 611134, Chinahualun@sicau.edu.cn (L.H.)

**Keywords:** FGF21, overiectomy, central obesity, corticosterone, insulin, metabolic syndromes

## Abstract

Increased glucocorticoid (GC) levels act as a master contributor to central obesity in estrogen-depleted females; however, what factors cause their increased GC production is unclear. Given (1) liver fibroblast growth factor 21 (FGF21) and GCs regulate each other’s production in a feed-forward loop, and (2) circulating FGF21 and GCs are parallelly increased in menopausal women and ovariectomized mice, we thus hypothesized that elevation of hepatic FGF21 secretion causes increased GGs production in estrogen-depleted females. Using the ovariectomized mice as a model for menopausal women, we found that ovariectomy (OVX) increased circulating corticosterone levels, which in turn increased visceral adipose *Hsd11b1* expression, thus causing visceral obesity in females. In contrast, liver-specific FGF21 knockout (FGF21 LKO) completely reversed OVX-induced high GCs and high visceral adipose *Hsd11b1* expression, thus abrogating OVX-induced obesity in females. Even though FGF21 LKO failed to rescue OVX-induced dyslipidemia, hepatic steatosis, and insulin resistance. What’s worse, FGF21 LKO even further exacerbated whole-body glucose metabolic dysfunction as evidenced by more impaired glucose and pyruvate tolerance and worsened insulin resistance. Mechanically, we found that FGF21 LKO reduced circulating insulin levels, thus causing the dissociation between decreased central obesity and the improvement of obesity-related metabolic syndromes in OVX mice. Collectively, our results suggest that liver FGF21 plays an essential role in mediating OVX-induced central obesity by promoting GC production. However, lack of liver FGF21 signaling reduces insulin production and in turn causes the dissociation between decreased central obesity and the improvement of obesity-related metabolic syndromes, highlighting a detrimental role for hepatic FGF21 signals in mediating the development of central obesity but a beneficial role in preventing metabolic abnormality from further exacerbation in estrogen-depleted females.

## 1. Introduction

Menopause in women usually causes central obesity, leading to the development of obesity-related abnormalities, including dyslipidemia, hepatic steatosis, hyperglycemia, insulin resistance, and cardiovascular diseases [1]. It is well known that menopause-associated hormonal changes are the predominant factors that cause central obesity and related metabolic abnormalities in women [2]. Generally, loss of systemic estrogens has long been considered to be the primary mediator of central obesity and related metabolic abnormalities in menopausal women [1]. However, recent studies using mouse models evidenced that the elevation of glucocorticoids (GCs; cortisol in humans and corticosterone in rodents) is the primary contributor to central obesity and related metabolic abnormalities in estrogen-depleted females [3]. Because deprivation of GCs by adrenalectomy could completely prevent or reverse ovariectomy (OVX)-induced central obesity and related lipid metabolic abnormalities (e.g., hepatic steatosis) in female mice [3]. Further, the sharp increase of follicle-stimulating hormone (FSH) has also been confirmed to be another key causative factor for the development of central obesity as well as related metabolic abnormalities in menopausal women and OVX mouse models [3,4,5]. Therefore, the mechanisms underpinning menopause-induced central obesity and related metabolic abnormalities have not been fully explained so far. Even though it is clear that multiple altered endocrine hormones should act together to synergistically contribute to the development of central obesity and related metabolic abnormalities in menopausal women, thus, elucidation of the complex crosstalk between these altered endocrine hormones in the process of menopause-induced central obesity will provide key insights into such metabolic syndromes as well as better treatment.

Liver-derived fibroblast growth factor 21 (FGF21) is a recently recognized metabolic regulator for both carbohydrates and lipids, exerting great roles in maintaining glucose, lipid, and energy homeostasis [6]. Amounting studies documented that, along with increased GCs, serum FGF21 levels were dramatically increased in both menopausal women [7] and ovariectomized mice [8] as well. Further, the parallel increase in circulating FGF21 and GCs levels was also observed in obese objects, including both rodents and humans [9,10,11,12], revealing that FGF21 may also play a key role in the development of obesity. Intriguingly, it has been evidenced that FGF21 and GCs regulate each other’s production in a feed-forward loop [13]. Accordingly, we hypothesize that the increase of circulating FGF21 is an important causative factor for the development of central obesity in estrogen-depleted females via promoting GC production.

In this study, using OVX mice as a model, we first reported that liver-specific FGF21 knockout (FGF21 LKO) could completely reverse the elevation of circulating GCs and thus abrogate OVX-induced central obesity in mice. However, FGF21 LKO also caused the dissociation between decreased central obesity and the improvement of obesity-related metabolic syndromes, highlighting a complex role for hepatic FGF21 in the metabolic regulation of estrogen-depleted females. Our novel findings provide new insights into the etiology of central obesity in estrogen-depleted females and, meanwhile, are also crucial in developing effective prevention and management strategies for central obesity in menopausal women.

## 2. Results

### 2.1. FGF21 LKO Abrogates OVX-Induced Central Obesity in Female Mice

Circulating FGF21 is mainly liver-derived and plays important roles in regulating lipid homeostasis and body weight balance [14]. To explore whether liver-derived FGF21 is involved in OVX-induced body weight gain and obesity in females, we bilaterally ovariectomized FGF21 LKO and their wild-type (WT) littermates at 6 weeks of age and then monitored their body weight change weekly until the age of 30 weeks. Resultantly, OVX obviously increased (*p* < 0.05) body weight of female mice two weeks after surgery procedures, and then their body weight was persistently maintained higher (*p* < 0.05) than that of the sham control females to the end of the experiment (Figure 1A). In stark contrast, the body weight of OVX+FGF21 LKO mice was persistently maintained at a comparable level (*p* > 0.05) to the sham control females, both of which were substantially lower (*p* < 0.05) than that of the OVX mice (Figure 1A), suggesting that FGF21 LKO completely abrogated OVX-induced body weight gain in females.

Using ELISA, we found that OVX drastically increased (*p* < 0.001) circulating FGF21 levels, while FGF21 LKO reduced (*p* < 0.01) circulating FGF21 levels to very low levels in female mice (Figure 1B). At decapitation, the visceral adipose weight of OVX mice was largely increased (*p* < 0.001), while FGF21 LKO completely abrogated (*p* < 0.001) OVX-induced visceral adipose accumulation in females (Figure 1C,D). Additionally, the visceral adipose mass in OVX+FGF21 LKO mice was even numerically lower than in sham controls (Figure 1D), suggesting FGF21 LKO caused a lipodystrophy of visceral adipose in mice. Histology analysis indicated that the adipocyte size of visceral adipose was comparable between and OVX+FGF21 LKO mice and OVX mice, both of which was obviously larger than that of the sham control mice (Figure 1E). It is well known that FGF21 serves as a key anti-obesogenic factor [9]. Thus, the increase of circulating FGF21 was functionally contradictory to the observed increase in visceral adipose deposition in OVX mice, and vice versa in OVX+FGF21 LKO mice, implicating that the abrogative effects of FGF21 LKO on OVX-induced central obesity should be realized using mechanisms rather than lipolytic action of FGF21 directly in adipose tissues.

### 2.2. FGF21 LKO Fails to Rescue OVX-Caused Dyslipidemia and Hepatic Steatosis in Mice

Dyslipidemia and hepatic steatosis are usually the hallmarks of menopausal obesity. As expected, compared to the sham control females, OVX substantially increased (*p* < 0.01) circulating triglyceride (TG) (Figure 2A) and free fatty acid (FFA) (Figure 2B) levels in mice. However, FGF21 LKO failed to rescue OVX-induced dyslipidemia in females, as both TG and FFA levels were comparable (*p* > 0.05) between OVX+FGF21 LKO and OVX mice (Figure 2A,B). Even though FGF21 LKO still mildly decreased the circulating TG levels in OVX mice to comparable levels (*p* > 0.05) of the sham controls (Figure 2A).

Compared to the sham controls, OVX increased (*p* < 0.01) liver weight (Figure 2C) and liver TG content (Figure 2D) in mice. Liver tissue H&E staining also verified that OVX increased fat deposition in the liver and caused hepatic steatosis in female mice (Figure 2E), while FGF21 LKO failed to reverse (*p* > 0.05) the OVX-induced liver weight gain and hepatic steatosis in OVX mice (Figure 2C–E). Except steatosis, no other obvious pathomorphological changes were observed in the liver tissues from either OVX or OVX+FGF21 LKO mice.

### 2.3. FGF21 LKO Exacerbates Glucose Metabolic Abnormalities in OVX Mice

Except for causing obesity, OVX also causes hyperglycemia and raises the risk of insulin resistance and even diabetes in females [1], while liver-derived FGF21 is a pivotal factor to regulate both glucose metabolism and insulin sensitivity [14]. To see whether liver-derived FGF21 is involved in glucose metabolic dysfunction in OVX mice, we performed glucose (GTT), insulin (ITT) and pyruvate tolerance (PTT) tests. As expected, compared to the sham controls, OVX increased (*p* < 0.05) fasting blood glucose levels (Figure 3B), and increased insulin (*p* = 0.08) and pyruvate (*p* < 0.01) intolerance in mice, indicating that OVX decreased insulin sensitivity and impaired liver glucose production capacity. Surprisingly, the combination of OVX and FGF21 LKO resulted in exacerbation of whole body glucose metabolic abnormalities, as reflected by more impaired glucose and pyruvate tolerance, and worsened (*p* = 0.09) insulin resistance in OVX female mice after FGF21 LKO (Figure 3A–F). These results revealed that liver-derived FGF21 plays important roles in protecting the whole body glucose metabolic abnormalities from further exacerbation in OVX mice.

### 2.4. Transcriptomic Profiling Highlighting 11β-HSD1 Plays a Central Role in Mediating FGF21 LKO on Abrogating OVX-Induced Central Obesity

To further understand the underlying mechanisms by which FGF21 LKO abrogated OVX-induced central obesity in female mice, we profiled the transcriptome of visceral adipose tissues. Pairwise comparison among groups revealed 1186 differentially expressed genes (DEGs, Figure 4A; see ‘Section 5’ for criteria). Of those, 1083 DEGs were identified between OVX and sham control mice, of which 591 (54.6%) were upregulated and 492 (45.4%) downregulated in OVX mice (Figure 4B). Some of the well-known obesogenic or obesity marker genes, such as *Hsd11b1* and *Lep,* were obviously upregulated by OVX, while some of the well-known fatty acid oxidation genes, e.g., *Ucp1* were downregulated by OVX (Appendix A). Functional enrichment analysis using DAVID (v2023q2) indicated that OVX induced fat cell differentiation and intracellular lipid transport, thus promoting visceral fat deposition; meanwhile, OVX-induced excess fat deposition also dampened angiogenesis, neurogenesis, and tissue morphogenesis of visceral adipose(Appendix A), thus resulting in inflammation in visceral adipose tissues.

Strikingly, once specifically knocking out FGF21 in the liver, the expression of 928 (86%) out of 1083 DEGs in visceral adipose of OVX females was recovered back to the sham control levels, which we refer to as recovered DEGs (rDEGs, Figure 4C, and Appendix A). The expression of massive DEGs (86%) in OVX mice was recovered back to the sham control levels after FGF21 LKO, strongly suggesting a pivotal role of liver-derived FGF21 in OVX-induced obesity. Functional enrichment analysis of these rDEGs indicated that FGF21 LKO recovered angiogenesis, neurogenesis, cell proliferation/morphogenesis, cell adhesion, and cell surface receptor signaling pathway in visceral adipose tissues of OVX mice (Appendix A). Of those rDEGs, a total of 77 were involved in the lipid metabolic process (GO:0006629) (Figure 4D). Among the 77 rDEGs involved in the lipid metabolic process, only *HSD11b1*, *Apod,* and *Dkk3* were DEGs between OVX+FGF21 LKO and OVX. 11β-hydroxysteroid dehydrogenase type 1 (*11β-HSD1*), encoded by *HSD11b1*, coverts inactive 11-dehydrocorticosterone to active corticosterone in rodents (cortisone and cortisol in humans, respectively). Previous studies have evidenced that 11β-HSD1 acts as the essential regulator of GCs excess to cause central obesity [15], highlighting that 11β-HSD1 may play a central role in mediating FGF21 LKO on abrogating OVX-induced obesity.

Gene set enrichment analysis (GSEA) using all the detected genes between OVX+FGF21 LKO versus OVX mice indicated that FGF21 LKO reduced adipogenesis in visceral adipose tissues of OVX mice (Figure 4E). Additionally, further leading edge gene analysis also highlighted that *Hsd11b1* plays a major role in mediating FGF21 LKO on abrogating OVX-induced visceral obesity (Figure 4E). Taken together, all these analyses highlighted the central role of 11β-HSD1 in mediating FGF21 LKO on abrogating OVX-induced visceral obesity.

### 2.5. FGF21 LKO Reverses OVX-Induced High Corticosterone but Not High FSH

The increased secretion of follicle-stimulating hormone (FSH) and GCs following estrogen deprivation have been evidenced to be two pivotal factors that contribute to central obesity, dyslipidemia, insulin resistance, and hepatic steatosis in both menopausal women and OVX mice [3]. To see whether the abrogative effects of FGF21 LKO on OVX-induced central obesity were realized via altering FSH and/or corticosterone levels in female mice, we quantified their concentrations in serum using ELISA. Consequently, both serum FSH (Figure 5A) and corticosterone (Figure 5B) levels were substantially increased (*p* < 0.05) in mice following OVX. In stark contrast, FGF21 LKO completely reversed (*p* < 0.01) high circulating corticosterone but not high FSH in OVX mice (Figure 5A,B), suggesting the abrogative effect of FGF21 LKO on OVX-induced central obesity was realized via reversing corticosterone production but not FSH in females.

To further ascertain whether the decreased circulating corticosterone in OVX+FGF21 LKO mice was due to the drop of circulating FGF21, we administrated OVX+FGF21 LKO mice with recombinant mouse FGF21 and then sampled blood for serum corticosterone quantification. Consequently, transient recombinant FGF21 replacement could largely increase (*p* < 0.01) corticosterone levels in OVX+FGF21 LKO mice (Figure 5B), reinforcing liver-derived FGF21 exerts an essential role in mediating OVX-induced high corticosterone production in females. 11β-HSD1 expression and activities are regulated by GC excess in a feed-forward loop [15]. Previous studies have evidenced that abrogating circulating GCs by adrenalectomy could completely prevent OVX-induced central obesity in mice [3], and 11β-HSD1 knockout also could prevent GC excess-induced central obesity [15]. Using qPCR, we found that in response to increased corticosterone, the mRNA expression of *Hsd11b1* in visceral adipose tissues of OVX+FGF21 LKO mice was largely increased following recombinant FGF21 replacement as well (Figure 5C). Collectively, these results suggested that OVX increased corticosterone production in estrogen-depleted females in a FGF21-dependent manner. The increased corticosterone, in turn, triggered adipose local *Hsd11b1* expression, thereby leading to central obesity in estrogen-depleted females.

### 2.6. FGF21 LKO Causes Hypoinsulinemia in OVX Mice

Menopause in women or OVX in mice usually causes hyperinsulinemia due to insulin resistance [16]. We found that OVX caused higher (*p* = 0.1) serum insulin levels in mice when compared to the sham control (Figure 6A), while FGF21 LKO drastically reduced (*p* < 0.001) serum insulin levels in OVX mice (Figure 6A). Unexpectedly, the serum insulin levels in OVX+FGF21 LKO mice were even lower (*p* < 0.001) than that of the sham controls (Figure 6A).

Given insulin exerts great roles in stimulating adipogenesis and lipogenesis via enhancing expression of key adipogenic transcription factors, e.g., SREBF1 [17], we thus performed GSEA analysis of all the detected genes between OVX+FGF21 LKO and sham control mice to check whether adipogenesis in OVX+FGF21 LKO mice was also impaired in relative to the sham controls. Consequently, as expected, Srebf1-mediated de novo fatty acid synthesis in visceral adipose was impaired (FDR = 0.028) in OVX+FGF21 LKO mice when compared to the sham controls (Figure 6B). Leading edge gene analysis verified that Srebf1 targeting de novo fatty acid synthesis genes, including *Acaca*, *Elovl6*, *Fasn,* and *Scd,* exert a major role in mediating FGF21 LKO on abrogating adipogenesis in visceral depot (Figure 6B). These data suggest that the drop of visceral adipose deposition in OVX+FGF21 LKO mice was associated, at least partially, with the decrease in insulin secretion as well. Further, we also found that insulin signaling in visceral adipose tissue was dampened or impaired (FDR = 0.037) by FGF21 LKO in OVX mice when compared to the sham controls (Figure 6C). Further, leading-edge gene analysis revealed that *Irs1* plays a major role in mediating FGF21 LKO on dampening insulin signaling in visceral adipose tissues.

### 2.7. The Potential Mechanism and Key Genes by Which FGF21 LKO Abrogates OVX-Induced Central Obesity in Mice

According to the above analyses, we could conclude that FGF21 LKO abrogated OVX-induced central obesity mainly by reversing high corticosterone levels and reducing insulin production in OVX mice (Figure 7A). Using protein–protein interaction (PPI) network analysis, we found that the rDEGs involved in the lipid metabolism process formed a complex network centered on glucocorticoid receptor (*Nr3c1*) and insulin receptor substrate 1 (*Irs1*) (Figure 7B), reinforcing the fact that FGF21 LKO abrogates OVX-induced central obesity via modulating both GC/GR signaling and insulin signaling. These rDEGs shown in Figure 7B should play important roles in mediating FGF21 LKO on abrogating OVX-induced obesity.

## 3. Discussion

The prevalence of central obesity and related metabolic abnormalities have become a major threat to the health and lifespan of menopausal women [18]. Due to the complexity of its pathologies, so far, there is still no effective strategy to control the incidence and development of central obesity as well as related metabolic abnormalities in menopausal women [18]. In this study, we first found that liver-specific FGF21 knockout could completely abrogate OVX-induced central obesity by reversing high GC production in mice, highlighting an essential role of hepatic FGF21 in mediating the development of central obesity in estrogen-depleted females.

Adrenal gland-secreted GCs act as a master factor for controlling nutrient substrate availability and systemic fuel partitioning, thereby essentially regulating adipose expansion, distribution, and whole-body energy homeostasis [19]. In humans, excessive production of GCs causes Cushing’s syndrome, characterized by central obesity, while insufficient production of GCs causes Addison’s disease with minimal adipose deposition [20]. Both menopausal women [21] and OVX mice [3] develop Cushing-like syndrome characterized by drastically increased circulating GCs levels and the development of central obesity. Intriguingly, abrogating high circulating GCs by adrenalectomy could completely prevent or reverse OVX-induced central obesity in mice [3], highlighting the essential role of GCs in mediating the development of central obesity in estrogen-depleted females. However, what factors cause increased GCs in estrogen-depleted females is still unclear so far. In this study, we found that circulating FGF21 and GCs were parallelly increased in female mice following OVX treatment and depletion of circulating FGF21 by FGF21 LKO completely reversed high circulating GCs, thus abrogating OVX-induced central obesity in female mice. Furthermore, transient replacement of recombinant FGF21 to OVX+FGF21 LKO mice could completely rescue OVX-induced high circulating GCs. All these results strongly demonstrate that liver-derived FGF21 plays an essential role in mediating OVX-induced central obesity by promoting high GC production.

In addition to the regulated production of GCs by the hypothalamic-pituitary-adrenal (HPA) axis, the abundance of active GCs can be modulated by local tissue enzymes. 11Beta-hydroxysteroid dehydrogenase type 1 (11β-HSD1) converts inactive cortisone (in humans) or 11-dehydrocorticosterone (in rodents) into its active form cortisol (in humans) or corticosterone (in rodents) [22] thereby regulating the local availability of active GCs. Previous studies indicated that GCs excess acts as a feed-forward signal to increase tissue local GCs availability by stimulating 11β-HSD1 expression and activities, and global 11β-HSD1 knockout mice were protected from GCs excess-caused central obesity [15], suggesting an essential role for 11β-HSD1 in mediating circulating GCs excess for causing visceral adipose accumulation and the incidence of central obesity. In the present study, we found FGF21 LKO completely abrogated OVX-induced *11β-HSD1* hyper-expression in visceral adipose tissues, and further replacement of recombinant FGF21 to the OVX+FGF21 LKO mice could significantly rescue their visceral adipose *11β-HSD1* hyper-expression. Accordingly, we suggested that the liver FGF21-adrenal GCs-visceral adipose *11β-HSD1* signaling cascade plays a central role in mediating OVX-induced central obesity. Thus, strategies targeting any element in such a signaling cascade may be applied to prevent and treat central obesity in estrogen-depleted females.

It is well known that insulin plays a crucial role in both adipogenesis and lipogenesis, and the deletion of the constitutive insulin receptor (IR) in adipocytes causes lipodystrophy [23]. Intriguingly, recent studies newly evidenced that GCs excess causes hyperinsulinemia, which is a primary factor inducing central obesity [24]. In the present study, OVX increased circulating GCs and caused hyperinsulinemia, while FGF21 LKO completely reversed OVX-induced high GCs and caused hypoinsulinemia in mice. Therefore, the drop in GCs likely decreased insulin levels, which may be another important mechanism that mediates FGF21 LKO to abrogate OVX-induced central obesity in females. Further, FGF21 itself also plays a crucial role in preserving β-cell function and survival and in the maintenance of normal islet cell growth, islet function, and insulin synthesis [25]. Thus, in addition to decreasing GCs’s production, deficiency of FGF21 per se also should have direct effects to reduce insulin production/secretion and, in turn, visceral adipose deposition in OVX+FGF21 LKO mice. This was possibly why the serum insulin levels and visceral fat weight of OVX+FGF21 LKO mice were even significantly lower or tended to be lower than those of the sham controls.

The incidence of central obesity usually causes metabolic syndromes, such as dyslipidemia, hepatic steatosis, and insulin resistance [26]. Consistently, all these metabolic syndromes occurred in female mice following OVX in the present study. Unexpectedly, FGF21 LKO completely abrogated OVX-induced central obesity but not these obesity-related metabolic disorders. Thus, it appears that the increased visceral adipose accumulation is not the major or determinant factor for causing these metabolic syndromes, or in other words, there is a dissociation between pure increased visceral adipose deposition and the incidence of metabolic disorders in estrogen-depleted females. Indeed, in both menopausal women and OVX mice, multiple factors or mechanisms have been verified to be involved in their metabolic deterioration [1,3]. For example, the drastic elevation of FSH has been well established to exert great roles in causing dyslipidemia, hepatic steatosis, and insulin resistance in both menopausal women and OVX mice [3,5,27]. However, serum FSH levels were not reversed by FGF21 LKO in OVX mice. Collectively, our results suggest that the altered endocrine hormones like increased FSH rather than mere increased visceral adipose deposition may play a more important role in promoting the development of metabolic abnormalities, including dyslipidemia, hepatic steatosis, and insulin resistance in estrogen-depleted females.

Surprisingly, FGF21 LKO did not alleviate but even exacerbated whole body glucose metabolic abnormalities in OVX mice, as evidenced by more impaired glucose and pyruvate tolerance, and worsened insulin resistance in OVX+FGF21 LKO mice compared to OVX mice. Integrated data analysis also indicated that FGF21 LKO impaired insulin signaling in OVX mice. Insulin plays a central role in regulating whole-body glucose homeostasis [17]. In the present study, FGF21 LKO drastically decreased the circulating insulin levels in OVX mice, which could well explain why the metabolic abnormality in OVX mice was further exacerbated by FGF21 KO. Furthermore, in addition to acting as a master factor for modulating glucose homeostasis, insulin is also essential for the formation, maintenance, and function of white adipose tissue [17]. Deficiency of insulin signaling permits unregulated adipose tissue lipolysis, leading to dyslipidemia and the ectopic deposition of lipids in non-adipose tissues such as the liver, further causing hepatic steatosis and exacerbating systemic insulin resistance [28]. Thus, reduced insulin production may be the central factor responsible for the dissociation between decreased visceral adipose deposition and the improvement of metabolic abnormalities in OVX+FGF21 LKO mice. Accordingly, mild suppression of hyper-FGF21 without unfavorable insulin-associated metabolic changes should be an interesting strategy to prevent or treat OVX/menopause-caused central obesity.

In summary, liver-derived FGF21 plays an essential role in mediating the development of central obesity in OVX mice by promoting GC production. However, FGF21 LKO also reduced circulating insulin levels, leading to the dissociation between decreased visceral adipose deposition and the improvement of metabolic abnormalities, highlighting the detrimental role of hepatic FGF21 signaling in the development of central obesity but a beneficial role in protecting metabolic abnormality from further exacerbation in estrogen-depleted females. Our novel findings provide new insights into the pathology of central obesity and related metabolic disorders in estrogen-depleted females and will facilitate the development of new strategies for preventing or treating central obesity and related metabolic abnormalities in menopausal women.

## 4. Limitation of the Study

Although the result of the present study provides new insights into the etiology of central obesity in estrogen-depleted females, there are still some limitations in this study. Firstly, we did not explore how FGF21 LKO reduced corticosterone production. So, the molecular mechanisms by which FGF21 LKO reverses OVX-induced central obesity still require further elucidation. Secondly, how FGF21 LKO reduced circulating insulin levels in OVX mice is also unknown and warrants further studies. Thirdly, our conclusion was drawn based on studies conducted in OVX mice; whether the same is true in women is unknown and requires further validation. Finally, despite the beneficial effects of FGF21 LKO on preventing OVX-induced central obesity, it also impaired insulin production/signaling and exacerbated glucose metabolic dysregulation. Thus, how to prevent or treat OVX/menopause-induced central obesity and meanwhile rescue metabolic abnormalities is still required further investigation.

## 5. Materials and Methods

### 5.1. Animals and Treatments

The liver-specific FGF21 knockout mice were generated as in our previous descriptions [29]. Namely, the FGF21^loxp/loxp^ mice (Jackson Laboratory; Stock # 022361; Bar Harbor, ME, USA) were crossed to Alb^Cre/Cre^ mice (Jackson Laboratory; Stock # 003574; Bar Harbor, ME, USA) to generate Alb^Cre/^; FGF21^loxp/−^ mice. Then, the Alb^Cre/^; FGF21^loxp/−^ mice were intercrossed to generate Alb^Cre+^; FGF21^loxp/loxp^ mice (FGF21 LKO) and Alb^Cre−^; FGF21^loxp/loxp^ mice (WT) for this study. For studying the effects of liver-derived FGF21 on OVX-induced central obesity in female mice, 12 FGF21 LKO and 24 WT female littermates were selected. At 6 weeks of age, 12 FGF21 LKO (OVX+FGF21 LKO) and 12 randomly selected WT (OVX) were ovariectomized under anesthesia. To clarify the effects of OVX and OVX+FGF21 LKO on adipose deposition and related metabolic abnormalities, the remaining 12 WT littermates were undergone sham surgery and used as the control. After surgery, all mice were maintained in a controlled microenvironment with a temperature of 21 ± 2 °C, a relative humidity of 50–60%, and a 12 h light/12 h dark cycle (lights on at 06:00 a.m.) as measured using the facility monitoring system (Phoenix Controls, Newtown Square, PA, and Comdale Systems, Toronto, Ontario, Canada). All mice had free access to water and a standard chow. Additionally, lighting conditions were controlled using a programmable electronic timer (ChronTrol XT, ChronTrol Corp., San Diego, CA, USA). Mice were weighted weekly until sacrifice at age of 30 weeks. The experimental protocol was approved by the Ethics Review Committee for Animal Experimentation of Sichuan Agricultural University (No: 20220612). All animal experiments were conducted in accordance with the institutional guidelines for laboratory animals established by the animal care and use committee of Sichuan Agricultural University, and all methods strictly obeyed the Guide for the ARRIVE (Animal Research: Reporting of In Vivo Experiments) guidelines 2.0 [30].

### 5.2. Glucose, Insulin and Pyruvate Tolerance Tests

One to two week(s) before decapitation, all mice have undergone glucose, insulin, and pyruvate tolerance tests according to previous studies with minor modifications [31]. Namely, for the glucose tolerance test (GTT), all mice were fasting overnight for 18 h, then intraperitoneally (i.p.) injected with D-glucose (Sigma-Aldrich, St. Louis, MI, USA) at a dose of 1 g/kg body weight. Blood glucose concentrations were measured at 0, 15, 30, 60, 90, and 120 min with a handheld glucometer (Roche Diagnostics, Mannheim, Germany). For the insulin tolerance test (ITT) and pyruvate tolerance test (PTT), the mice were i.p. injected with the regular 30/70 recombinant human insulin (Eli Lily, Indianapolis, IN, USA) at a dose of 0.75 U/kg body weight after a 4 h fasting, or pyruvate (Sigma-Aldrich) at a dose of 1.5 mg/kg body weight after a 18 h fasting. Then, blood glucose concentrations were measured at 0, 15, 30, 45, and 60 min for both tests with a handheld glucometer (Roche Diagnostics). Areas under the curve (AUCs) were calculated for glucose, insulin and pyruvate tolerance tests using the blood glucose data.

### 5.3. Sample Collections and Parameter Measurements at Decapitation

At 30 weeks of age, all mice were anesthetized with isoflurane (Fluriso; VetOne, Boise, ID, USA), weighed and euthanized. For decreasing the influence of estrus cyclicity on experimental results, all mice from the sham group were sacrificed at the dioestrus phase. The trunk blood was collected and centrifuged at 2000× *g* for 15 min at 4 °C, and the sera were stored at −20 °C, pending analyses. After sacrifice, the abdominal adipose tissues were collected and weighed, one portion of the abdominal adipose tissues were immediately frozen in liquid nitrogen and stored at −80 °C for RNA-sequencing analysis, and the remaining abdominal adipose tissues were fixed in 10% buffered formalin and used for histology analysis.

### 5.4. Serum Hormone and Lipid Determination

Serum FGF21 (MF2100; R&D Systems, Minneapolis, MN, USA), FSH (CSB-E06871m; CUSABIO, Houston, TX, USA), corticosterone (KGE009; R&D Systems, Minneapolis, MN, USA) and insulin (80-INSHU-CH01; ALPCO, Salem, NH, USA) concentrations were determined with commercial ELISA kits, according to the manufacturer’s instruction. Serum-free fat acid (FFA) and triglyceride (TG) levels were measured via a nonesterified free fatty acids assay kit (A042-2-1) and triglyceride assay Kit (A110-1-1) from Nanjing Jiancheng Bioengineering Institute (Nanjing, China), as our previous descriptions [32]. The liver tissue of mice was added with nine times the amount of saline to prepare 10% tissue homogenization in the ice water bath, and TG levels in the liver tissue of mice were measured using the enzymic method according to the kit instructions.

### 5.5. RNA-Sequencing

Total RNA was extracted from the crushed visceral adipose tissue samples using TRIzol™ Reagent (Invitrogen, Carlsbad, CA, USA). RNA quantity and quality were determined by using the RNA Nano 6000 Assay Kit of the Bioanalyzer 2100 system (Agilent Technologies, Palo Alto, CA, USA). 1000 ng of total RNA per sample with RNA integrity numbers (RINs) greater than or equal to 8.0 was used as input material to prepare the RNA sample. Briefly, mRNA was purified from total RNA by using oligo (dT) magnetic capture beads. Libraries were synthesized by using the NEBNext UltraTM RNA Library Prep Kit for Illumina (NEB, Ipswich, MA, USA) according to the manufacturer’s instructions. Then, the cDNA library was sequenced on an illu-mina Novaseq platform and generated 150 bp paired-end reads. Fastp (Version 0.19.7) was used to filter low-quality reads (defined as reads with more than 50% beads scoring Qphred ≤ 20) and to remove adapter sequences to generate clean reads. Mouse reference genome (http://ftp.ensembl.org/pub/release-105/fasta/mus_musculus/dna/, accessed on 16 May 2022) and gene annotation files (http://ftp.ensembl.org/pub/release-105/gtf/mus_musculus/, accessed on 16 May 2022) were down-loaded from Ensembl directly to build the index of the reference genome; then clean reads were then aligned to the reference genome using Hisat2 (v2.0.5). For gene expression level quantification, featureCounts (v1.5.0-p3) was used to count the reads numbers mapped to each gene. The R package DESeq2 (1.20.0) was used to perform pairwise comparisons among groups; the resulting *p*-values were adjusted using the Benjamini and Hochberg’s approach for controlling the false discovery rate (FDR), the gene met the criteria of FDR < 0.05 were considered as a differentially expressed gene (DEG).

### 5.6. Functional Enrichment Analysis

Database for Annotation, Visualization and Integrated Discovery (DAVID) bioinformatics database (v2023q2) was used to perform gene ontology (GO) and Kyoto Encyclopedia of Genes and Genomes (KEGG) analysis. For all GO terms and KEGG pathways, a threshold of *p* value < 0.05 was set for significance.

### 5.7. Protein–Protein Interaction Network Analysis

Protein–protein interaction (PPI) network was built up using the STRING APP of Cytoscape (3.9.1); the minimum required interaction score was set as 0.4. Cytoscape (3.9.1) was used for network visualization.

### 5.8. FGF21 Administration In Vivo

To evaluate whether FGF21 replacement could rescue OVX-induced high circulating corticosterone levels in OVX+FGF21 LKO mice, eight wild-types (Alb^Cre-^; FGF21^loxp/loxp^) and 16 FGF21 LKO (Alb^Cre+^; FGF21^loxp/loxp^) female mice from the breeding cages were selected, and all mice were bilaterally ovariectomized at 6 wk of age, and randomly allocated into two subgroups (n = 8). Four weeks after surgery procedures (i.e., at 10 week of age), a subgroup of OVX+FGF21 LKO mice were injected via tail vein with recombinant mouse FGF21 (RD272108100, BioVendor, Prague, Czech Republic) at a dose of 1 mg/kg body weight at 0800 AM, according to our previous descriptions [29]. Additionally, the other subgroup of OVX+FGF21 LKO and OVX mice were injected with vehicle (0.9% saline). Two hours after injection, all mice were decapitated, the trunk blood was collected, and visceral adipose was obtained and stored at −80 °C.

### 5.9. Quantitative Reverse Transcription PCR (qRT-PCR)

To verify the effects of FGF21 replacement on visceral adipose *Hsd11b1* expression, the visceral adipose tissues from animals were collected for RT-qPCR analysis. Briefly, total RNA was isolated from visceral adipose tissues using TRIZOL(Invitrogen Co., Carlsbad, CA, USA), according to the manufacturer’s instructions. Quantitative and qualitative analyses of isolated RNA were assessed from the ratio of absorbance at 260 and 280 nm and agarose gel electrophoresis. First-strand complementary DNA (cDNA) was reverse-transcribed (500 ng total RNA) using a PrimeScript^®^ RT reagent kit with gDNA Eraser (TaKaRa Bio, Co., Ltd., Dalian, China) and used as the template for subsequent real-time quantitative PCR reaction (RT-qPCR), done in triplicate on a CFX96 Real-Time PCR detection system (Bio-Rad, Hercules, CA, USA) with SYBR^®^ greenII. RT-qPCR was conducted for 43 cycles with 5 s at 94 °C for denaturing, 25 s at 60 °C for annealing and primer extension, and a final melting curve analysis to monitor the purity of the PCR product. Each PCR reaction (10 μL) contained the same amount of cDNA (1 μL template), 500 nmol/L each of forward and reversed primers, and 2XSYBR^®^ premix Taq^TM^ (TaKaRa Bio Co., Ltd., Beijing, China). The cycle threshold value was analyzed (CFX96 detection system) and transformed to a relative quantity using a standard curve. Relative gene expression levels were normalized to *GAPDH*. Outcomes were expressed as fold changes relative to average mRNA levels of genes in sham controls. The sequences of primers used for qRT-PCR are as follows: *Hsd11b1* forward: 5′- CAGAAATGCTCCAGGGAAAGAA-3′, reverse: 5′- GCAGTCAATACCACATGGGC-3′; *GAPDH* forward: 5′- AGGTCGGTGTGAACGGATTTG-3′, reverse: 5′- TGTAGACCATGTAGTTGAGGTCA-3′.

### 5.10. Statistical Analyses

All statistical analyses except RNA-sequencing data were performed with GraphPad Prism 9.2 software (La Jolla, CA, USA). Comparisons among groups were carried out via one-way ANOVA followed by Turkey’s test. For analysis of the effect of treatment in repeated measures (i.e., body weight), two-way ANOVA followed by Sidak’s multiple comparisons test was used. All values were presented as mean ± SEM. Significance is given as * *p* < 0.05, ** *p* < 0.01 and *** *p* < 0.001.

## Figures and Tables

**Figure 1 ijms-24-14922-f001:**
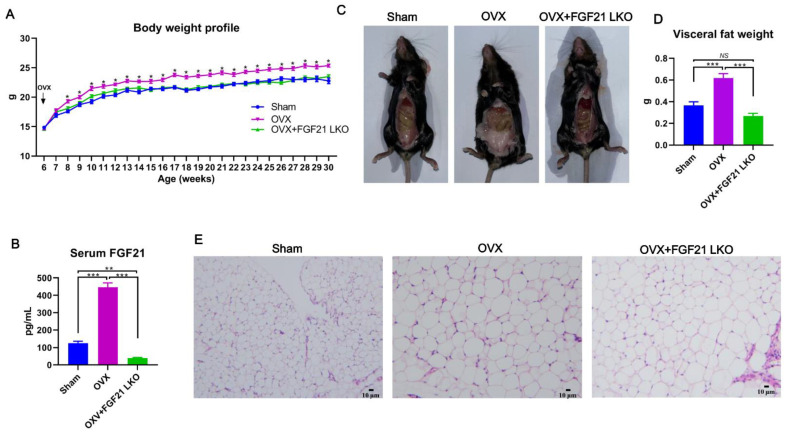
Liver-specific FGF21 knockout abrogated OVX-induced central obesity in mice. (**A**) Body weight profile of mice following OVX or the combination of OVX plus liver-specific FGF21 knockout. (**B**) The serum FGF21 concentrations in mice at decapitation. (**C**) Representative figures of mice indicating visceral adipose deposition at decapitation. (**D**) The average visceral fat weight of mice from each group. (**E**) Representative H&E staining of visceral adipose tissues of mice at decapitation. In figure A: * denotes *p* < 0.05 (OVX versus Sham/OVX+FGF21 LKO); in figure B and D: ** *p* < 0.01; *** *p* < 0.001, ^NS^ *p* > 0.05.

**Figure 2 ijms-24-14922-f002:**
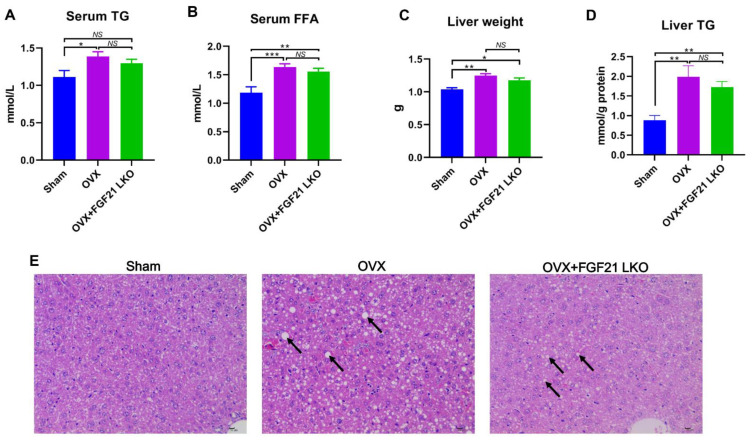
Liver-specific FGF21 knockout failed to rescue OVX-induced dyslipidemia and hepatic steatosis in mice. (**A**) Serum concentration of triglyceride (TG). (**B**) Serum concentration of free fat acid (FFA). (**C**) Liver weight. (**D**) Liver TG content. (**E**) The representative H&E staining of liver tissues from each group. Arrows indicate lipid droplets. * *p* < 0.05; ** *p* < 0.01; *** *p* < 0.001; ^NS^ *p* > 0.05.

**Figure 3 ijms-24-14922-f003:**
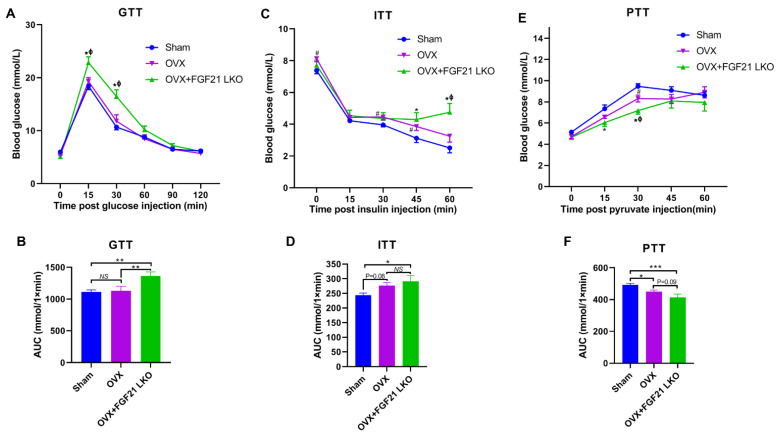
Liver-specific FGF21 knockout exacerbated OVX-induced glucose metabolic abnormalities in mice. (**A**) Glucose tolerance test (GTT). (**B**) The area under the curve of GTT. (**C**) Insulin tolerance test (ITT). (**D**) The area under the curve of ITT. (**E**) Pyruvate tolerance test (PTT). (**F**) The area under the curve of PTT. Note figure (**A**,**C**,**E**): * OVX+FGF21 LKO versus Sham (*p* < 0.05), ^ϕ^ OVX+FGF21 LKO versus OVX (*p* < 0.05), ^#^ OVX versus Sham (*p* < 0.05); figure (**B**,**D**,**F**): * *p* < 0.05; ** *p* < 0.01; ****p* < 0.001; ^NS^ *p* > 0.05.

**Figure 4 ijms-24-14922-f004:**
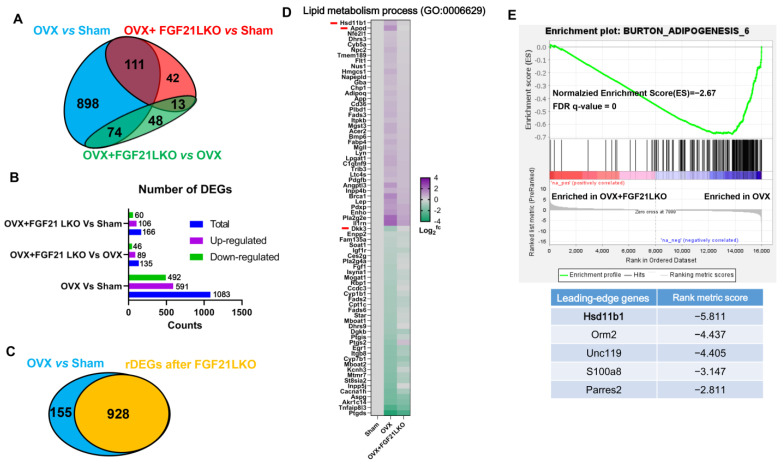
Transcriptomic profiling revealing *Hsd11b1* plays a central role in mediating FGF21 LKO on abrogating OVX-induced central obesity in mice. (**A**) Venn diagram of differentially expressed genes (DEGs) among groups. (**B**) The number of DEGs between pairwise groups. (**C**) The recovered DEGs (rDEGs) in OVX mice following FGF21 LKO. rDEGs are genes whose expression in OVX was significantly different from the sham controls but recovered back to the sham levels after FGF21 LKO. (**D**) The heat map of rDEGs involved in lipid metabolism process (GO:0006629). Red lines indicate DEGs between OVX+FGF21 LKO versus OVX. (**E**) Gene set enrichment analysis (GSEA) showed FGF21 LKO reduced adipogenesis in OVX mice, and leading-edge gene analysis highlighted that *Hsd11b1* plays a major role in mediating FGF21 LKO on preventing adipogenesis in OVX mice.

**Figure 5 ijms-24-14922-f005:**
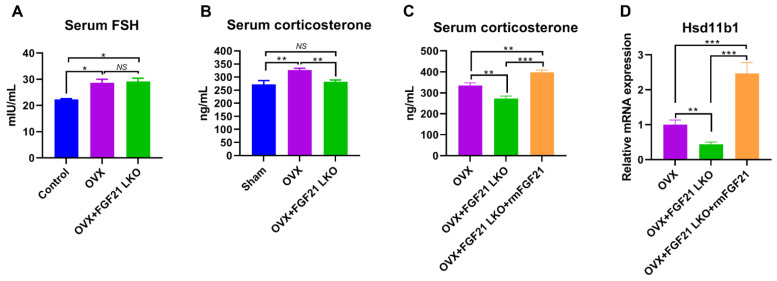
FGF21 LKO reversed circulating high corticosterone but not high FSH in OVX mice. (**A**) Serum concentration of FSH. (**B**) Serum concentration of corticosterone. (**C**) Effects of transient recombinant FGF21 replacement on serum concentration of corticosterone in OVX+FGF21 LKO mice. (**D**) Effects of transient recombinant FGF21 replacement on visceral adipose Hsd11b1 expression in OVX+FGF21 LKO mice. * *p* < 0.05; ** *p* < 0.01; *** *p* < 0.001; ^NS^ *p* > 0.05.

**Figure 6 ijms-24-14922-f006:**
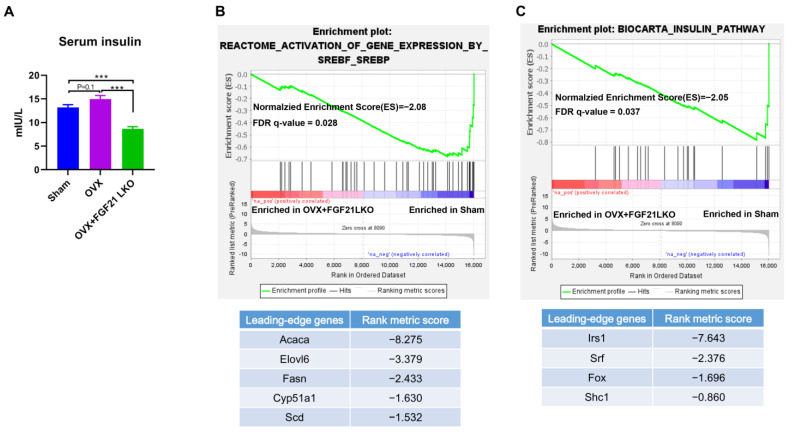
FGF21 LKO reduced serum insulin levels in OVX mice. (**A**) Serum concentration of insulin; (**B**) Gene set enrichment analysis indicated FGF21 LKO reduced SREBF-mediated lipogenesis. (**C**) Gene set enrichment analysis indicated FGF21 LKO even reduced insulin signaling in OVX mice compared to the sham controls. *** *p* < 0.001.

**Figure 7 ijms-24-14922-f007:**
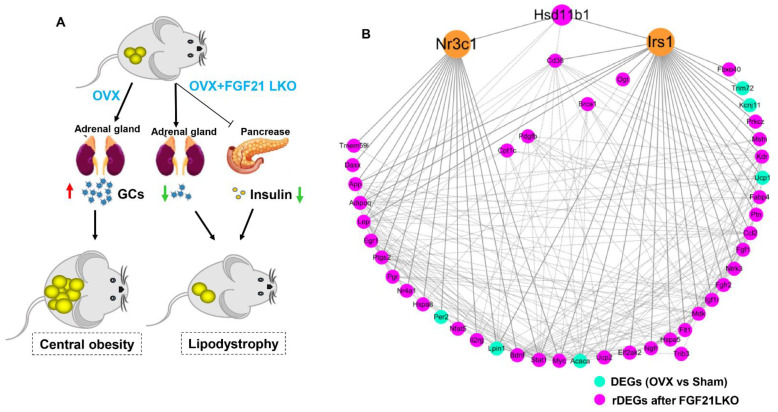
The potential mechanism and key genes by which FGF21 LKO abrogated OVX-induced central obesity in mice. (**A**) The potential mechanism by which FGF21 LKO abrogated OVX-induced central obesity in mice. Liver-specific FGF21 knockout reduced both GC and insulin production, which in turn decreased adipogenesis and lipogenesis in visceral adipose tissues. (**B**) The potential key genes mediating FGF21 LKO on abrogating OVX-induced central obesity in mice.

## Data Availability

All data generated or analyzed during this study are included in this published article and its Appendix A.

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
