# Peer review of "Hepatic-Specific FGF21 Knockout Abrogates Ovariectomy-Induced Obesity by Reversing Corticosterone Production"

_ijms, 2023, doi:10.3390/ijms241914922_

Round 1

Reviewer 1 Report

The manuscript “Hepatic-Specific FGF21 Knockout Abrogates Ovariectomy-Induced Obesity by Reversing Corticosterone Production” shows the relationship between glucocorticoid production, obesity. and FGF21.

Their results are promising because liver-specific FGF21 knockout reverses elevated GCs and elevated Hsd11b1 expression in visceral adipose using OVX mice as a model. However, FGF21 LKO failed to rescue dyslipidemia and hepatic steatosis, which are often the hallmarks of menopausal obesity and cause detrimental metabolic syndromes and diseases. Surprisingly, FGF21 LKO did not alleviate but even exacerbated whole-body glucose metabolic abnormalities in OVX mice, as evidenced by the further impairment of glucose and pyruvate tolerance and worsening of insulin resistance in OVX+FGF21 LKO mice compared to OVX mice.

The authors have analyzed FGF21 LKO mainly in terms of DEGs involved in obesogenic genes or obesity markers, such as Hsd11b1 and Lep, and fatty acid oxidation genes, e.g., Ucp1. However, I cannot appreciate this extensive analysis of significant consequences of obesity such as insulin resistance, hyperglycemia, and pyruvate tolerance that are not reversed by FGF21 LKO.

And, finally, the authors should propose a solution to the exacerbated whole-body glucose metabolic abnormalities in postmenopausal women in addition to FGF21 LKO.

Author Response

Responds to the comments of Reviewer #1:

General comments: The manuscript “Hepatic-Specific FGF21 Knockout Abrogates Ovariectomy-Induced Obesity by Reversing Corticosterone Production” shows the relationship between glucocorticoid production, obesity. and FGF21. Their results are promising because liver-specific FGF21 knockout reverses elevated GCs and elevated Hsd11b1 expression in visceral adipose using OVX mice as a model. However, FGF21 LKO failed to rescue dyslipidemia and hepatic steatosis, which are often the hallmarks of menopausal obesity and cause detrimental metabolic syndromes and diseases. Surprisingly, FGF21 LKO did not alleviate but even exacerbated whole-body glucose metabolic abnormalities in OVX mice, as evidenced by the further impairment of glucose and pyruvate tolerance and worsening of insulin resistance in OVX+FGF21 LKO mice compared to OVX mice. The authors have analyzed FGF21 LKO mainly in terms of DEGs involved in obesogenic genes or obesity markers, such as Hsd11b1 and Lep, and fatty acid oxidation genes, e.g., Ucp1. However, I cannot appreciate this extensive analysis of significant consequences of obesity such as insulin resistance, hyperglycemia, and pyruvate tolerance that are not reversed by FGF21 LKO.

Response: We appreciated the Reviewer’s comments on our work.

Specific comments: Finally, the authors should propose a solution to the exacerbated whole-body glucose metabolic abnormalities in postmenopausal women in addition to FGF21 LKO.

Response: Thanks for suggestions. According to our results, FGF21 plays a detrimental role to promote central obesity but also exerts a beneficial role to protect metabolic abnormalities, especially glucose metabolic dysregulation from further exacerbation in estrogen-depleted females. Thus the key solution to the exacerbated whole-body glucose metabolic abnormalities is deceasing the circulating FGF21 at an appropriate extent. In our study, liver-specific FGF21 knockout resulted in very low (depleted) circulating FGF21 in mice, thus causing exacerbated whole-body glucose metabolic abnormalities. So, if not depleting but just mildly decreasing circulating FGF21 using pharmaceutical regents without unfavorable insulin-associated metabolic changes may not only prevent OVX/menopause-induced central obesity but also rescue metabolic abnormalities. The above solution has been added in the Discussion section (marked in red). Additionally, we also analyzed the limitations of the study and proposed the potential future study directions to resolve estrogen deficiency-caused metabolic abnormalities (see Section 5; marked in red).

 Special thanks to your good comments and suggestions!

Reviewer 2 Report

Title:  Hepatic-Specific FGF21 Knockout Abrogates Ovariectomy-In-duced Obesity by Reversing Corticosterone Production

Authors: Jiayu Xu, Xinyu Shao, Haozhe Zeng, Chengxi wang, Jiayi Li, Xiaoqin Peng, Yong Zhuo, Lun Hua, Fengyan Meng, Xingfa Han

General comment:

In a world struggling with an obesity pandemic, there is a need to study the pathomechanisms leading to the development of this disease, which may provide new therapeutic targets in the future. In their study, Jiayu Xu et al. investigated the effect of FGF21 gene liver-specific knockout on the development of obesity and metabolic complications in ovariectomized menopausal model mice. They found that the absence of the gene encoding FGF21 inhibits weight gain in ovariectomized mice, but this does not translate into metabolic benefits. The research hypothesis is well-founded and the methodology is appropriately chosen. Nevertheless, there are several issues that the authors should address before the manuscript is accepted for publication.

Minor revisions:

- Authors: I suggest replacing the first letter in the author's name Chengxi wang with a capital letter;

- Abstract and introduction: please correct a typo: “liver fibroblast growth factor 12 (FGF21)”;

- Abstract: please add information on transcriptomic profiling;

- Results: Section 2.7 requires the title to be replaced as the current one is a repetition of the title from section 2.6.;

- Figure 1C: Please consider changing “Reprehensive” to “Representative”;

- Discussion: Please consider possible limitations of the study;

- Whole manuscript: please explain the abbreviations as only they occur in the text;

- Whole manuscript: the manuscript may benefit from careful editing.

The manuscript may benefit from the assistance of a native English speaker and careful editing.

Author Response

Responds to the comments of Reviewer #2:

General comments: In a world struggling with an obesity pandemic, there is a need to study the pathomechanisms leading to the development of this disease, which may provide new therapeutic targets in the future. In their study, Jiayu Xu et al. investigated the effect of FGF21 gene liver-specific knockout on the development of obesity and metabolic complications in ovariectomized menopausal model mice. They found that the absence of the gene encoding FGF21 inhibits weight gain in ovariectomized mice, but this does not translate into metabolic benefits. The research hypothesis is well-founded and the methodology is appropriately chosen. Nevertheless, there are several issues that the authors should address before the manuscript is accepted for publication.

Response: We appreciated the Reviewer’s positive comments on our work. 

Specific comments:

Q1: Authors: I suggest replacing the first letter in the author's name Chengxi wang with a capital letter.

Response: Has been corrected. Sorry for this error.

Q2: Abstract and introduction: please correct a typo: “liver fibroblast growth factor 12 (FGF21)”.

Response: Such errors have been corrected across the manuscript. Thanks!

Q3: Abstract: please add information on transcriptomic profiling.

Response: Through transcriptomic profiling, we identified visceral Hsd11b1 played a key role in mediating OVX-induced central obesity, and meanwhile in mediating FGF21 LKO on abrogating OVX-induced central obesity. These two key points have been included in the Abstract. The other results obtained from transcriptomic profiling were not so important, so we did not added in the Abstract due to the word limits.  

Q4: Results: Section 2.7 requires the title to be replaced as the current one is a repetition of the title from section 2.6.

Response: Has been corrected. Thanks!

Q5: Figure 1C: Please consider changing “Reprehensive” to “Representative”;

Response: Has corrected to “Representative”. Thanks!

Q6: Discussion: Please consider possible limitations of the study.

Response: According to the Review’s suggestion, we have added a section of limitation of the study in the revised manuscript (Section 5; marked in red).

Q7: Whole manuscript: please explain the abbreviations as only they occur in the manuscript

Response: We have carefully checked the abbreviations and deleted those that are not present in the manuscript and added those that are present in the manuscript.

Q8: Whole manuscript: the manuscript may benefit from careful editing.

Response: We have carefully check the writing across the manuscript and also invited a language expert to edit our manuscript. Thanks for suggestion.

Special thanks to your good comments and suggestions!

Reviewer 3 Report

Comments to the Authors of manuscript ID: ijms-2620931 entitled “Hepatic-Specific FGF21 Knockout Abrogates Ovariectomy-Induced Obesity by Reversing Corticosterone Production”.

The study investigates the factors contributing to increased glucocorticoid (GC) levels, which are linked to central obesity in estrogen-depleted females. The researchers focused on the role of liver fibroblast growth factor 12 (FGF21) in this process, as FGF21 and GCs have a reciprocal influence on each other. They used ovariectomized (OVX) mice as a model for menopausal women and made several key observations:

OVX led to elevated corticosterone levels and increased expression of Hsd11b1 in visceral adipose tissue, resulting in visceral obesity in female mice.

Liver-specific knockout of FGF21 (FGF21 LKO) reversed OVX-induced high GC levels and adipose Hsd11b1 expression, preventing obesity in OVX females.

However, FGF21 LKO did not rescue OVX-induced dyslipidemia, hepatic steatosis, or insulin resistance and even worsened glucose metabolic dysfunction.

FGF21 LKO reduced circulating insulin levels, contributing to the dissociation between reduced central obesity and improved metabolic syndromes in OVX mice.

In summary, liver-derived FGF21 plays a crucial role in promoting GC production and central obesity in estrogen-depleted females. However, the absence of liver FGF21 signaling reduces insulin production, leading to a disconnect between reduced central obesity and the amelioration of metabolic abnormalities. This study highlights the complex interplay between FGF21, GCs, and insulin in the development of central obesity and related metabolic issues in menopausal women.

1. The introduction discusses the factors contributing to central obesity and related metabolic abnormalities in menopausal women. While the overall content seems informative, there are some grammatical and structural issues in the text.

2. The paper lacks a clear research question or hypothesis statement at the beginning. Clearly stating the research question or hypothesis would help readers understand the study's purpose from the outset.

3. the results appears to contain some grammatical and stylistic issues.

4. Moreover, results it is not discussion and the comparison of the data with other studies is wrong, e.g.: “Using ELISA, consistent with previous studies [8], we validated that OVX drastically”; “metabolic syndromes and diseases [1].’; “body weight balance [14]”. The results should be rephrased.

5. Figure 1 and 2 missed scalbars.

6. Figure 2. Lipid drops are small spherical structures that store excess fat in the liver cells. They can be stained with special dyes to visualize them under a microscope. One common stain is Oil Red O, which binds to neutral lipids and gives them a red color. Another stain is Nile Red. This staining presented is wrong and cannot be accepted.

7. There is a lack of pathomorphological description of the liver structure. It should be included.

8. please, avoid the phrase like: “we performed Transcriptomic profiling..”. such constructions should be rephrased.

9.  The discussion refers to various findings and results without citing specific studies or providing appropriate references. To strengthen the merit of the research, it's crucial to provide citations for each claim and statement, allowing readers to verify the information and connect it to existing literature.

10. The discussion could benefit from a more structured discussion. It jumps between various aspects of the study, making it challenging to follow the logical flow of the research findings. Organizing the discussion into subsections based on key findings or themes would improve clarity.

11. While the text presents many results and observations, it could do a better job of interpreting these findings. What are the implications of the results? How do they contribute to the understanding of the research question? Clear explanations and interpretations would enhance the merit of the study.

12. The text is quite lengthy, which may make it challenging for readers to stay engaged. Consider condensing the information and focusing on the most critical points to maintain reader interest.

13. It's important to acknowledge the limitations of the study and suggest avenues for future research. Addressing potential limitations and proposing future directions would demonstrate a comprehensive understanding of the research's scope.

14. When presenting statistical results, it's important to indicate the level of significance (e.g., p-values) to help readers assess the strength of the findings.

15. why only WT were shame operated? Where are FGF21LKO? It should be explained or this group should be included into the study presented here.

16. The methods mentions the number of mice used in the study (12 FGF21 LKO and 24 WT), but it doesn't explain the rationale for this sample size or how mice were randomly assigned to experimental groups. Providing this information is essential for the statistical validity of the study.

17. While the methods briefly mentions that the experimental protocol was approved by an ethics committee and conducted following institutional guidelines, it could benefit from a more detailed explanation of how animal welfare and ethical considerations were addressed throughout the study. This includes aspects like anesthesia, monitoring, and humane endpoints.

18. The methods mentions the housing conditions (e.g., temperature, light/dark cycle) but doesn't provide details about how these conditions were controlled or monitored. Providing this information ensures the reproducibility of experiments conducted in similar conditions.

19. The methods mentions that mice were weighed weekly until sacrifice at the age of 30 weeks. However, it doesn't explain why this time frame was chosen or whether any other time points were considered during the study. The rationale for the experimental timeline should be clarified.

20. The methods doesn't describe the specific data collected during the study (e.g., measurements related to obesity) or the statistical methods used for data analysis. Providing this information is crucial for understanding how the results were obtained and interpreted.

21. While the methods mentions a control group (Sham), it doesn't explain the role of this group in the study or what specific comparisons were made between the experimental groups. Clarifying the purpose of the control group is important for understanding the experimental design.

22. material and methods:

a. Some methods and reagents are mentioned without specific references or sources. It's important to provide proper citations for these methods and reagents to allow readers to access detailed information if needed.

b. The text mentions the number of mice used in the study but doesn't explain the rationale behind the choice of sample size. It's essential to justify the sample size based on statistical considerations, such as power analysis or previous studies.

c. The Authors do not mention whether randomization and blinding were used during experiments. Randomization helps ensure that groups are comparable, while blinding reduces potential bias. Including information about these practices is important for experimental rigor.

d. Describing how RNA sequencing data were processed, normalized, and analyzed would be valuable for researchers interested in replicating or understanding the analysis.

e. While it mentions using DAVID for functional enrichment analysis, it would be helpful to specify which specific analyses (e.g., GO, KEGG) were performed and how significance thresholds were determined.

f. the part of “Glucose, insulin and pyruvate tolerance tests” should be supported with references

Author Response

Responds to the comments of Reviewer #3:

General Comment: The study investigates the factors contributing to increased glucocorticoid (GC) levels, which are linked to central obesity in estrogen-depleted females. The researchers focused on the role of liver fibroblast growth factor 12 (FGF21) in this process, as FGF21 and GCs have a reciprocal influence on each other. They used ovariectomized (OVX) mice as a model for menopausal women and made several key observations. OVX led to elevated corticosterone levels and increased expression of Hsd11b1 in visceral adipose tissue, resulting in visceral obesity in female mice. Liver-specific knockout of FGF21 (FGF21 LKO) reversed OVX-induced high GC levels and adipose Hsd11b1 expression, preventing obesity in OVX females. However, FGF21 LKO did not rescue OVX-induced dyslipidemia, hepatic steatosis, or insulin resistance and even worsened glucose metabolic dysfunction. FGF21 LKO reduced circulating insulin levels, contributing to the dissociation between reduced central obesity and improved metabolic syndromes in OVX mice. In summary, liver-derived FGF21 plays a crucial role in promoting GC production and central obesity in estrogen-depleted females. However, the absence of liver FGF21 signaling reduces insulin production, leading to a disconnect between reduced central obesity and the amelioration of metabolic abnormalities. This study highlights the complex interplay between FGF21, GCs, and insulin in the development of central obesity and related metabolic issues in menopausal women

Response: We appreciated the Reviewer’s positive comments on our work.

Specific comments:

Q1: The introduction discusses the factors contributing to central obesity and related metabolic abnormalities in menopausal women. While the overall content seems informative, there are some grammatical and structural issues in the manuscript.

Response: According to the Review’s comments, we have revised the Introduction to make it more informative. And, we carefully checked and corrected the grammatical and structural issues across the manuscript (marked in red).

Q2: The paper lacks a clear research question or hypothesis statement at the beginning. Clearly stating the research question or hypothesis would help readers understand the study's purpose from the outset.

Response: According to the review’s good comments, we added a hypothesis in the Induction section (“Accordingly, we hypothesize that the increase of circulating FGF21 is an important causative factor for the development of central obesity in estrogen-depleted females via promoting GCs production”; marked in red).

Q3: The results appears to contain some grammatical and stylistic issues.

Response: We have carefully checked and revised the grammatical and stylistic issues across the manuscript. And, we also invited a person to edit our manuscript.

Q4: Moreover, results it is not discussion and the comparison of the data with other studies is wrong, e.g.: “Using ELISA, consistent with previous studies [8], we validated that OVX drastically”; “metabolic syndromes and diseases [1].’; “body weight balance [14]”. The results should be rephrased.

Response: Thanks for suggestions, we have deleted or rephrased all of these “discussion and comparison of our data with other studies ” sentences in Result section. But, some references in the Result section were not used for discussion or comparison but for presenting results, so we still kept them. Citing some references to help describe the results is very common (PMID: 29514097, Cell Report, 2018). 

Q5: Figure 1 and 2 missed scalbars.

Response: Thanks for suggestion, we have added the scalbars.

Q6: Figure 2. Lipid drops are small spherical structures that store excess fat in the liver cells. They can be stained with special dyes to visualize them under a microscope. One common stain is Oil Red O, which binds to neutral lipids and gives them a red color. Another stain is Nile Red. This staining presented is wrong and cannot be accepted.

Response: We totally agree with the Reviewer's comments. Yes, Oil Red O or Nile Red staining is usually used to visualize the lipid drops in liver. But, when checking the published papers, we can see some studies using both H&E and Oil Red O staining (PMCID: PMC8604675; Redox Biol, 2021; PMID: 21459323, Cell Metab, 2011), and some studies only used H&E staining (PMID:19177596, Hepatology, 2009), to visualize lipid drops in liver. At the initial of the study, we focused on the OVX-induced central obesity, so we neglected this issue and only using H&E staining to check the liver lipid deposition. Now, we have no stored liver tissues to repeat Oil Red O or Nile Red staining. In our study, we have directly measured liver TG content, and evidenced OVX increased TG deposition in liver, whereas FGF21 LKO abrogated OVX-induced TG deposition in liver. Yes, we totally agreed that only H&E staining is not so good, but it still can help to shown the lipid deposition in liver (PMID:19177596, Hepatology, 2009). So, we still want to keep this result in the manuscript.

Q7: There is a lack of pathomorphological description of the liver structure. It should be included.

Response: According to suggestion, we added pathomorphological descriptions of the liver structure in the manuscript (marked in red).

Q8: Please, avoid the phrase like: “we performed Transcriptomic profiling..”. such constructions should be rephrased.

Response: All such type of phrases have been revised or rephrased in the manuscript. Thanks for suggestion,

Q9: The discussion refers to various findings and results without citing specific studies or providing appropriate references. To strengthen the merit of the research, it's crucial to provide citations for each claim and statement, allowing readers to verify the information and connect it to existing literature.

Response: Thanks for suggestions, we have checked and provided appropriate references.

Q10: The discussion could benefit from a more structured discussion. It jumps between various aspects of the study, making it challenging to follow the logical flow of the research findings. Organizing the discussion into subsections based on key findings or themes would improve clarity.

Response: According to the Review’s good suggestions, we have revised the Discussion structure based on our key findings to improve the clarity.

Q11: While the manuscript presents many results and observations, it could do a better job of interpreting these findings. What are the implications of the results? How do they contribute to the understanding of the research question? Clear explanations and interpretations would enhance the merit of the study.

Response: In the Discussion section, we added more interpretations, implications and explanations of our findings to enhance the merit of our study.

Q12: The manuscript is quite lengthy, which may make it challenging for readers to stay engaged. Consider condensing the information and focusing on the most critical points to maintain reader interest.

Response: Thanks for suggestion, we have tried our best to improve the discussion, deleted some contents and focused on the critical points of our studies to maintain reader interest.

Q13: It's important to acknowledge the limitations of the study and suggest avenues for future research. Addressing potential limitations and proposing future directions would demonstrate a comprehensive understanding of the research's scope.

Response: We have added a paragraph to discuss the limitation of the study and propose future directions (Section 5; marked in red).

Q14: When presenting statistical results, it's important to indicate the level of significance (e.g., p-values) to help readers assess the strength of the findings.

Response: We checked and added the p-values to the statistical findings in cases where they are missing.

Q15: why only WT were sham operated? Where are FGF21LKO? It should be explained or this group should be included into the study presented here.

Response: Because FGF21 LKO mice were also ovariectomized, so only WT mice needed to be sham operated.

Q16: The methods mentions the number of mice used in the study (12 FGF21 LKO and 24 WT), but it doesn't explain the rationale for this sample size or how mice were randomly assigned to experimental groups. Providing this information is essential for the statistical validity of the study.

Response: Both FGF21 LKO and WT (AlbCre-; FGF21loxp/loxp) sibling mice were selected and used in the studies. Considering the big differences between individual organisms (mice), we usually used a litter big sample size (i.e., sample size 12 ) to increase statistical power in our previous studies. So, we also used 12 mice within a group in this study. The 24 WT mice were randomly allocated into OVX or Sham groups according to the computer-generated random sequence of numbers. Statistical analysis indicated that there was no difference (p > 0.05) in bodyweight at the onset of the study (Fig. 1A). These information has been added in the manuscript.

Q17: While the methods briefly mentions that the experimental protocol was approved by an ethics committee and conducted following institutional guidelines, it could benefit from a more detailed explanation of how animal welfare and ethical considerations were addressed throughout the study. This includes aspects like anesthesia, monitoring, and humane endpoints.

Response: All animal procedures in this study were strictly obeyed the Guide for the ARRIVE (Animal Research: Reporting of in Vivo Experiments) guidelines 2.0 (PMID: 32663221). So, details regarding anesthesia, monitoring, and humane endpoints all could find in the Guide for the ARRIVE(PMID: 32663221). We have added these information into the manuscript, and cited this reference.

Q18: The methods mentions the housing conditions (e.g., temperature, light/dark cycle) but doesn't provide details about how these conditions were controlled or monitored. Providing this information ensures the reproducibility of experiments conducted in similar conditions.

Response: Have provided details about how to control these conditions in the manuscript (Marked in red).

Q19: The methods mentions that mice were weighed weekly until sacrifice at the age of 30 weeks. However, it doesn't explain why this time frame was chosen or whether any other time points were considered during the study. The rationale for the experimental timeline should be clarified.

Response: Dear Reviewer, we had no special considerations, we just wanted to weekly monitor the bodyweight change of mice following OVX or OVX+FGF21 LKO. So, we think once a week is enough and good.

Q20: The methods doesn't describe the specific data collected during the study (e.g., measurements related to obesity) or the statistical methods used for data analysis. Providing this information is crucial for understanding how the results were obtained and interpreted.

Response: We have described these informations in Section 5.3. Usually, body weight and visceral adipose mass are used to indicate OVX-induced obesity. So, in this study we also use the two parameters to measure OVX-induced obesity. And, the statistical methods for data analysis are seen in Section 5.10.

Q21: While the methods mentions a control group (Sham), it doesn't explain the role of this group in the study or what specific comparisons were made between the experimental groups. Clarifying the purpose of the control group is important for understanding the experimental design.

Response: Performing sham surgery on the control mice was to remove negative effects of the surgery stress on experimental outcomes. The control (Sham) group was used to verify whether OVX induces obesity, and whether liver-specific FGF21 knockout could reverse OVX-induced obesity. In our study, all multiple comparisons were conducted among the three groups (OVX, OVX+FGF21 LKO and Sham). According to suggestion, we have clarify the purpose of designing the sham group in the manuscript (marked in red).  

Q22: material and methods: a. Some methods and reagents are mentioned without specific references or sources. It's important to provide proper citations for these methods and reagents to allow readers to access detailed information if needed.

Response: We have checked and added specific references or sources to methods and reagents used in this study.

Q23: material and methods: b. The manuscript mentions the number of mice used in the study but doesn't explain the rationale behind the choice of sample size. It's essential to justify the sample size based on statistical considerations, such as power analysis or previous studies.

Response: Have explained above.

Q24: material and methods: c. The Authors do not mention whether randomization and blinding were used during experiments. Randomization helps ensure that groups are comparable, while blinding reduces potential bias. Including information about these practices is important for experimental rigor.

Response: Thanks for suggestions. In both the two experiments of the study, mice were randomly allocated into different group, which have been clarified in the revised manuscript.

Q25: material and methods: d. Describing how RNA sequencing data were processed, normalized, and analyzed would be valuable for researchers interested in replicating or understanding the analysis.

Response: We have clearly described how the RNA-sequencing data was processed, normalized and analyzed (see Section 5.5; marked in red).

Q26: material and methods: e. While it mentions using DAVID for functional enrichment analysis, it would be helpful to specify which specific analyses (e.g., GO, KEGG) were performed and how significance thresholds were determined.

Response: All these details have been clearly specified in Section 5.6 (marked in red).

Q27: material and methods: f. the part of “Glucose, insulin and pyruvate tolerance tests” should be supported with references

Response: According to suggestion, we have added references.

Special thanks to your good comments!

Round 2

Reviewer 1 Report

The authors have solved my concerns and I now consider that the manuscript can be published

Reviewer 3 Report

I have no more comments